# Bandit Convex Optimization: Towards Tight Bounds

**Elad Hazan**
Technion—Israel Institute of Technology
Haifa 32000, Israel
ehazan@ie.technion.ac.il

**Kfir Y. Levy**
Technion—Israel Institute of Technology
Haifa 32000, Israel
kfiryl@tx.technion.ac.il

## Abstract

Bandit Convex Optimization (BCO) is a fundamental framework for decision making under uncertainty, which generalizes many problems from the realm of online and statistical learning. While the special case of linear cost functions is well understood, a gap on the attainable regret for BCO with *nonlinear* losses remains an important open question. In this paper we take a step towards understanding the best attainable regret bounds for BCO: we give an efficient and near-optimal regret algorithm for BCO with strongly-convex and smooth loss functions. In contrast to previous works on BCO that use time invariant exploration schemes, our method employs an exploration scheme that shrinks with time.

## 1 Introduction

The power of Online Convex Optimization (OCO) framework is in its ability to generalize many problems from the realm of online and statistical learning, and supply universal tools to solving them. Extensive investigation throughout the last decade has yield efficient algorithms with worst case guarantees. This has lead many practitioners to embrace the OCO framework in modeling and solving real world problems.

One of the greatest challenges in OCO is finding tight bounds to the problem of Bandit Convex Optimization (BCO). In this "bandit" setting the learner observes the loss function only at the point that she has chosen. Hence, the learner has to balance between exploiting the information she has gathered and between exploring the new data. The seminal work of [5] elegantly resolves this "exploration-exploitation" dilemma by devising a combined explore-exploit gradient descent algorithm. They obtain a bound of $\mathcal{O}(T^{3/4})$ on the expected regret for the general case of an adversary playing bounded and Lipschitz-continuous convex losses.

In this paper we investigate the BCO setting assuming that the adversary is limited to inflicting strongly-convex and smooth losses and the player may choose points from a *constrained* decision set. In this setting we devise an efficient algorithm that achieves a regret of $\tilde{O}(\sqrt{T})$. This rate is the best possible up to logarithmic factors as implied by a recent work of [11], cleverly obtaining a lower bound of $\Omega(\sqrt{T})$ for the same setting.

During our analysis, we develop a full-information algorithm that takes advantage of the strong-convexity of loss functions and uses a self-concordant barrier as a regularization term. This algorithm enables us to perform "shrinking exploration" which is a key ingredient in our BCO algorithm. Conversely, all previous works on BCO use a time invariant exploration scheme.

This paper is organized as follows. In Section 2 we introduce our setting and review necessary preliminaries regarding self-concordant barriers. In Section 3 we discuss schemes to perform single-

| Setting | Convex | Linear | Smooth | Str.-Convex | Str.-Convex & Smooth |
|---|---|---|---|---|---|
| Full-Info. | $\Theta(\sqrt{T})$ | | | $\Theta(\log T)$ | |
| BCO | $\tilde{O}(T^{3/4})$ | $\tilde{O}(\sqrt{T})$ | $\tilde{O}(T^{2/3})$ | | $\tilde{O}(\sqrt{T})$ [Thm. 10] |
| | $\Omega(\sqrt{T})$ | | | | |

Table 1: Known regret bounds in the Full-Info./ BCO setting. Our new result is highlighted, and improves upon the previous $\tilde{O}(T^{2/3})$ bound.

point gradient estimations, then we define first-order online methods and analyze the performance of such methods receiving noisy gradient estimates. Our main result is described and analyzed in Section 4; Section 5 concludes.

## 1.1 Prior work

For BCO with general convex loss functions, almost simultaneously to [5], a bound of $\mathcal{O}(T^{3/4})$ was also obtained by [7] for the setting of Lipschitz-continuous convex losses. Conversely, the best known lower bound for this problem is $\Omega(\sqrt{T})$ proved for the easier full-information setting.

In case the adversary is limited to using linear losses, it can be shown that the player does not "pay" for exploration; this property was used by [4] to devise the Geometric Hedge algorithm that achieves an optimal regret rate of $\tilde{O}(\sqrt{T})$. Later [1], inspired by interior point methods, devised the first *efficient* algorithm that attains the same nearly-optimal regret rate for this setup of bandit linear optimization.

For some special classes of nonlinear convex losses, there are several works that lean on ideas from [5] to achieve improved upper bounds for BCO. In the case of convex and smooth losses [9] attained an upper bound of $\tilde{O}(T^{2/3})$. The same regret rate of $\tilde{O}(T^{2/3})$ was achieved by [2] in the case of strongly-convex losses. For the special case of *unconstrained* BCO with strongly-convex and smooth losses, [2] obtained a regret of $\tilde{O}(\sqrt{T})$. A recent paper by Shamir [11], significantly advanced our understanding of BCO by devising a lower bound of $\Omega(\sqrt{T})$ for the setting of strongly-convex and smooth BCO. The latter implies the tightness of our bound.

A comprehensive survey by Bubeck and Cesa-Bianchi [3], provides a review of the bandit optimization literature in both stochastic and online setting.

## 2 Setting and Background

**Notation:** During this paper we denote by $\|\cdot\|$ the $\ell_2$ norm when referring to vectors, and use the same notation for the spectral norm when referring to matrices. We denote by $\mathbb{B}^n$ and $\mathbb{S}^n$ the $n$-dimensional euclidean unit ball and unit sphere, and by $v \sim \mathbb{B}^n$ and $u \sim \mathbb{S}^n$ random variables chosen uniformly from these sets. The symbol $\mathcal{I}$ is used for the identity matrix (its dimension will be clear from the context). For a positive definite matrix $A \succ 0$ we denote by $A^{1/2}$ the matrix $B$ such that $B^\top B = A$, and by $A^{-1/2}$ the inverse of $B$. Finally, we denote $[N] := \{1, \ldots, N\}$.

### 2.1 Bandit Convex Optimization

We consider a repeated game of $T$ rounds between a player and an adversary, at each round $t \in [T]$

1. player chooses a point $x_t \in \mathcal{K}$.

2. adversary independently chooses a loss function $f_t \in \mathcal{F}$.

3. player suffers a loss $f_t(x_t)$ and receives a feedback $\mathbb{F}_t$.

In the OCO (Online Convex Optimization) framework we assume that the decision set $\mathcal{K}$ is convex and that all functions in $\mathcal{F}$ are convex. Our paper focuses on adversaries limited to choosing functions from the set $\mathcal{F}_{\sigma,\beta}$; the set off all $\sigma$-strongly-convex and $\beta$-smooth functions.

We also limit ourselves to *oblivious* adversaries where the loss sequence $\{f_t\}_{t=1}^T$ is predetermined and is therefore independent of the player's choices. Mind that in this case the best point in hindsight is also independent of the player's choices. We also assume that the loss functions are defined over the entire space $\mathbb{R}^n$ and are strongly-convex and smooth there; yet the player may only choose points from a constrained set $\mathcal{K}$.

Let us define the regret of $\mathcal{A}$, and its regret with respect to a comparator $w \in \mathcal{K}$:

$$\text{Regret}_T^{\mathcal{A}} = \sum_{t=1}^T f_t(x_t) - \min_{w^* \in \mathcal{K}} \sum_{t=1}^T f_t(w^*), \qquad \text{Regret}_T^{\mathcal{A}}(w) = \sum_{t=1}^T f_t(x_t) - \sum_{t=1}^T f_t(w)$$

A player aims at minimizing his regret, and we are interested in players that ensure an $o(T)$ regret for any loss sequence that the adversary may choose.

The player learns through the feedback $\mathbb{F}_t$ received in response to his actions. In the full informations setting, he receives the loss function $f_t$ itself as a feedback, usually by means of a gradient oracle - i.e. the decision maker has access to the gradient of the loss function at any point in the decision set. Conversely, in the BCO setting the given feedback is $f_t(x_t)$, i.e., the loss function only at the point that he has chosen; and the player aims at minimizing his *expected regret*, $\mathbf{E}\left[\text{Regret}_T^{\mathcal{A}}\right]$.

## 2.2 Strong Convexity and Smoothness

As mentioned in the last subsection we consider an adversary limited to choosing loss functions from the set $\mathcal{F}_{\sigma,\beta}$, the set of $\sigma$-strongly convex and $\beta$-smooth functions, here we define these properties.

**Definition 1.** *(Strong Convexity) We say that a function $f : \mathbb{R}^n \to \mathbb{R}$ is $\sigma$-strongly convex over the set $\mathcal{K}$ if for all $x, y \in \mathcal{K}$ it holds that,*

$$f(y) \geq f(x) + \nabla f(x)^\top (y - x) + \frac{\sigma}{2} ||x - y||^2 \tag{1}$$

**Definition 2.** *(Smoothness) We say that a convex function $f : \mathbb{R}^n \to \mathbb{R}$ is $\beta$-smooth over the set $\mathcal{K}$ if the following holds:*

$$f(y) \leq f(x) + \nabla f(x)^\top (y - x) + \frac{\beta}{2} ||x - y||^2, \qquad \forall x, y \in \mathcal{K} \tag{2}$$

## 2.3 Self Concordant Barriers

Interior point methods are polynomial time algorithms to solving *constrained* convex optimization programs. The main tool in these methods is a *barrier function* that encodes the constrained set and enables the use of a fast *unconstrained* optimization machinery. More on this subject can be found in [8].

Let $\mathcal{K} \in \mathbb{R}^n$ be a convex set with a non empty interior $\text{int}(\mathcal{K})$
**Definition 3.** *A function $\mathcal{R} : int(\mathcal{K}) \to \mathbb{R}$ is called $\nu$-self-concordant if:*

1. *$\mathcal{R}$ is three times continuously differentiable and convex, and approaches infinity along any sequence of points approaching the boundary of $\mathcal{K}$.*

2. *For every $h \in \mathbb{R}^n$ and $x \in int(\mathcal{K})$ the following holds:*

$$|\nabla^3 \mathcal{R}(x)[h,h,h]| \leq 2(\nabla^2 \mathcal{R}(x)[h,h])^{3/2} \quad and \quad |\nabla \mathcal{R}(x)[h]| \leq \nu^{1/2}(\nabla^2 \mathcal{R}(x)[h,h])^{1/2}$$

here, $\nabla^3 \mathcal{R}(x)[h,h,h] := \frac{\partial^3}{\partial t_1 \partial t_2 \partial t_3} \mathcal{R}(x + t_1 h + t_2 h + t_3 h)\big|_{t_1=t_2=t_3=0}$.

Our algorithm requires a $\nu$-self-concordant barrier over $\mathcal{K}$, and its regret depends on $\sqrt{\nu}$. It is well known that any convex set in $\mathbb{R}^n$ admits a $\nu = \mathcal{O}(n)$ such barrier ($\nu$ might be much smaller), and that most interesting convex sets admit a self-concordant barrier that is efficiently represented.

The Hessian of a self-concordant barrier induces a local norm at every $x \in \text{int}(\mathcal{K})$, we denote this norm by $||\cdot||_x$ and its dual by $||\cdot||_x^*$ and define $\forall h \in \mathbb{R}^n$:

$$||h||_x = \sqrt{h^\top \nabla^2 \mathcal{R}(x) h}, \qquad ||h||_x^* = \sqrt{h^\top (\nabla^2 \mathcal{R}(x))^{-1} h}$$

we assume that $\nabla^2 \mathcal{R}(x)$ always has a full rank.

The following fact is a key ingredient in the sampling scheme of BCO algorithms [1, 9]. Let $\mathcal{R}$ is be self-concordant barrier and $x \in \text{int}(\mathcal{K})$ then the *Dikin Ellipsoide*,

$$W_1(x) := \{y \in \mathbb{R}^n : ||y - x||_x \leq 1\} \tag{3}$$

i.e. the $||\cdot||_x$-unit ball centered around $x$, is completely contained in $\mathcal{K}$.

Our regret analysis requires a bound on $\mathcal{R}(y) - \mathcal{R}(x)$; hence, we will find the following lemma useful:

**Lemma 4.** *Let $\mathcal{R}$ be a $\nu$-self-concordant function over $\mathcal{K}$, then:*

$$\mathcal{R}(y) - \mathcal{R}(x) \leq \nu \log \frac{1}{1 - \pi_x(y)}, \qquad \forall x, y \in int(\mathcal{K})$$

*where $\pi_x(y) = \inf\{t \geq 0 : x + t^{-1}(y - x) \in \mathcal{K}\}, \quad \forall x, y \in int(\mathcal{K})$*

Note that $\pi_x(y)$ is called the Minkowsky function and it is always in $[0, 1]$. Moreover, as $y$ approaches the boundary of $\mathcal{K}$ then $\pi_x(y) \to 1$.

## 3  Single Point Gradient Estimation and Noisy First-Order Methods

### 3.1  Single Point Gradient Estimation

A main component of BCO algorithms is a randomized sampling scheme for constructing gradient estimates. Here, we survey the previous schemes as well as the more general scheme that we use.

**Spherical estimators:**  Flaxman et al. [5] introduced a method that produces single point gradient estimates through spherical sampling. These estimates are then inserted into a full-information procedure that chooses the next decision point for the player. Interestingly, these gradient estimates are unbiased predictions for the gradients of a *smoothed version function* which we next define.

Let $\delta > 0$ and $v \sim \mathbb{B}^n$, the smoothed version of a function $f : \mathbb{R}^n \to \mathbb{R}$ is defined as follows:

$$\hat{f}(x) = \mathbf{E}[f(x + \delta v)] \tag{4}$$

The next lemma of [5] ties between the gradients of $\hat{f}$ and an estimate based on samples of $f$:

**Lemma 5.** *Let $u \sim \mathbb{S}^n$, and consider the smoothed version $\hat{f}$ defined in Equation (4), then the following applies:*

$$\nabla \hat{f}(x) = \mathbf{E}[\frac{n}{\delta} f(x + \delta u) u] \tag{5}$$

Therefore, $\frac{n}{\delta} f(x + \delta u) u$ is an unbiased estimator for the gradients of the smoothed version.

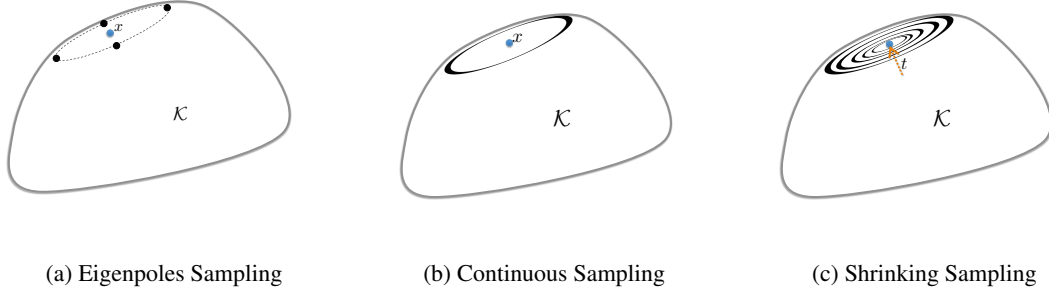

| (a) Eigenpoles Sampling | (b) Continuous Sampling | (c) Shrinking Sampling |

Figure 1: Dikin Ellipsoide Sampling Schemes

**Ellipsoidal estimators:** Abernethy et al. [1] introduced the idea of sampling from an ellipsoid (specifically the Dikin ellipsoid) rather than a sphere in the context of BCO. They restricted the sampling to the eigenpoles of the ellipsoid (Fig. 1a). A more general method of sampling continuously from an ellipsoid was introduced in [9] (Fig. 1b). We shall see later that our algorithm uses a "shrinking-sampling" scheme (Fig. 1c), which is crucial in achieving the $\tilde{O}(\sqrt{T})$ regret bound.

The following lemma of [9] shows that we can sample $f$ non uniformly over all directions and create an unbiased gradient estimate of a respective smoothed version:

**Corollary 6.** *Let* $f : \mathbb{R}^n \to \mathbb{R}$ *be a continuous function, let* $A \in \mathbb{R}^{n \times n}$ *be invertible, and* $v \sim \mathbb{B}^n$, $u \sim \mathbb{S}^n$. *Define the smoothed version of* $f$ *with respect to* $A$:

$$\hat{f}(x) = \mathbf{E}[f(x + Av)] \tag{6}$$

*Then the following holds:*

$$\nabla \hat{f}(x) = \mathbf{E}[n f(x + Au) A^{-1} u] \tag{7}$$

Note that if $A \succ 0$ then $\{Au : u \in \mathbb{S}^n\}$ is an ellipsoid's boundary.

Our next lemma shows that the smoothed version preserves the strong-convexity of $f$, and that we can measure the distance between $\hat{f}$ and $f$ using the spectral norm of $A^2$:

**Lemma 7.** *Consider a function* $f : \mathbb{R}^n \to \mathbb{R}$, *and a* positive definite *matrix* $A \in \mathbb{R}^{n \times n}$. *Let* $\hat{f}$ *be the smoothed version of* $f$ *with respect to* $A$ *as defined in Equation* (6)*. Then the following holds:*

- *If* $f$ *is* $\sigma$-*strongly convex then so is* $\hat{f}$.

- *If* $f$ *is convex and* $\beta$-*smooth, and* $\lambda_{\max}$ *be the largest eigenvalue of* $A$ *then:*

$$0 \le \hat{f}(x) - f(x) \le \frac{\beta}{2}||A^2||_2 = \frac{\beta}{2}\lambda_{\max}^2 \tag{8}$$

**Remark:** Lemma 7 also holds if we define the smoothed version of $f$ as $\hat{f}(x) = \mathbf{E}_{u \sim \mathbb{S}^n}[f(x + Au)]$ i.e. an average of the original function values over the unit sphere rather than the unit ball as defined in Equation (6). Proof is similar to the one of Lemma 7.

## 3.2 Noisy First-Order Methods

Our algorithm utilizes a full-information online algorithm, but instead of providing this method with exact gradient values we insert noisy estimates of the gradients. In what follows we define *first-order* online algorithms, and present a lemma that analyses the regret of such algorithm receiving noisy gradients.

**Definition 8.** *(First-Order Online Algorithm) Let $\mathcal{A}$ be an OCO algorithm receiving an arbitrary sequence of differential convex loss functions $f_1, \ldots, f_T$, and providing points $x_1 \leftarrow \mathcal{A}$ and $x_t \leftarrow \mathcal{A}(f_1, \ldots, f_{t-1})$. Given that $\mathcal{A}$ requires all loss functions to belong to some set $\mathcal{F}_0$. Then $\mathcal{A}$ is called* first-order online algorithm *if the following holds:*

- *Adding a linear function to a member of $\mathcal{F}_0$ remains in $\mathcal{F}_0$; i.e., for every $f \in \mathcal{F}_0$ and $a \in \mathbb{R}^n$ then also $f + a^\top x \in \mathcal{F}_0$*

- *The algorithm's choices depend only on its gradient values taken in the past choices of $\mathcal{A}$, i.e. :*
$$\mathcal{A}(f_1, \ldots, f_{t-1}) = \mathcal{A}(\nabla f_1(x_1), \ldots, \nabla f_{t-1}(x_{t-1})), \qquad \forall t \in [T]$$

The following is a generalization of Lemma 3.1 from [5]:

**Lemma 9.** *Let $w$ be a* fixed *point in $\mathcal{K}$. Let $\mathcal{A}$ be a first-order online algorithm receiving a sequence of differential convex loss functions $f_1, \ldots, f_T : \mathcal{K} \to \mathbb{R}$ ($f_{t+1}$ possibly depending on $z_1, \ldots z_t$). Where $z_1 \ldots z_T$ are defined as follows: $z_1 \leftarrow \mathcal{A}$, $z_t \leftarrow \mathcal{A}(g_1, \ldots, g_{t-1})$ where $g_t$'s are vector valued random variables such that:*

$$\mathbf{E}[g_t | z_1, f_1, \ldots, z_t, f_t] = \nabla f_t(z_t)$$

*Then if $\mathcal{A}$ ensures a regret bound of the form: $Regret_T^{\mathcal{A}} \leq B_{\mathcal{A}}(\nabla f_1(x_1), \ldots, \nabla f_T(x_T))$ in the full information case then, in the case of noisy gradients it ensures the following bound:*

$$\mathbf{E}[\sum_{t=1}^{T} f_t(z_t)] - \sum_{t=1}^{T} f_t(w) \leq \mathbf{E}[B_{\mathcal{A}}(g_1, \ldots, g_T)]$$

## 4 Main Result and Analysis

Following is the main theorem of this paper:

**Theorem 10.** *Let $\mathcal{K}$ be a convex set with diameter $\mathcal{D}_{\mathcal{K}}$ and $\mathcal{R}$ be a $\nu$-self-concordant barrier over $\mathcal{K}$. Then in the BCO setting where the adversary is limited to choosing $\beta$-smooth and $\sigma$-strongly-convex functions and $|f_t(x)| \leq L$, $\forall x \in \mathcal{K}$, then the expected regret of Algorithm 1 with $\eta = \sqrt{\frac{(\nu + 2\beta/\sigma) \log T}{2n^2 L^2 T}}$ is upper bounded as*

$$\mathbf{E}[Regret_T] \leq 4nL\sqrt{\left(\nu + \frac{2\beta}{\sigma}\right) T \log T} + 2L + \frac{\beta \mathcal{D}_{\mathcal{K}}^2}{2} = \mathcal{O}\left(\sqrt{\frac{\beta \nu}{\sigma} T \log T}\right)$$

*whenever $T/\log T \geq 2(\nu + 2\beta/\sigma)$.*

---

**Algorithm 1** BCO Algorithm for Str.-convex & Smooth losses

---

**Input**: $\eta > 0$, $\sigma > 0$, $\nu$-self-concordant barrier $\mathcal{R}$
Choose $x_1 = \arg\min_{x \in \mathcal{K}} \mathcal{R}(x)$
**for** $t = 1, 2 \ldots T$ **do**
  Define $B_t = \left(\nabla^2 \mathcal{R}(x_t) + \eta \sigma t \mathcal{I}\right)^{-1/2}$
  Draw $u \sim \mathbb{S}^n$
  Play $y_t = x_t + B_t u$
  Observe $f_t(x_t + B_t u)$ and define $g_t = n f_t(x_t + B_t u) B_t^{-1} u$
  Update $x_{t+1} = \arg\min_{x \in \mathcal{K}} \sum_{\tau=1}^{t} \left\{ g_\tau^\top x + \frac{\sigma}{2} ||x - x_\tau||^2 \right\} + \eta^{-1} \mathcal{R}(x)$
**end for**

---

Algorithm 1 shrinks the exploration magnitude with time (Fig. 1c); this is enabled thanks to the strong-convexity of the losses. It also updates according to a full-information first-order algorithm

denoted FTARL-$\sigma$, which is defined below. This algorithm is a variant of the FTRL methodology as defined in [6, 10].

---

**Algorithm 2** FTARL-$\sigma$

---

   **Input**: $\eta > 0$, $\nu$-self concordant barrier $\mathcal{R}$
   Choose $x_1 = \arg\min_{x \in \mathcal{K}} \mathcal{R}(x)$
   **for** $t = 1, 2 \dots T$ **do**
      Receive $\nabla h_t(x_t)$
      Output $x_{t+1} = \arg\min_{x \in \mathcal{K}} \sum_{\tau=1}^{t} \left\{ \nabla h_\tau(x_\tau)^\top x + \frac{\sigma}{2} ||x - x_\tau||^2 \right\} + \eta^{-1} \mathcal{R}(x)$
   **end for**

---

Next we give a proof sketch of Theorem 10

*Proof sketch of Therorem 10.* Let us decompose the expected regret of Algorithm 1 with respect to $w \in \mathcal{K}$:

$$\mathbf{E}\left[\text{Regret}_T(w)\right] := \sum_{t=1}^{T} \mathbf{E}\left[f_t(y_t) - f_t(w)\right]$$

$$= \sum_{t=1}^{T} \mathbf{E}\left[f_t(y_t) - f_t(x_t)\right] \tag{9}$$

$$+ \sum_{t=1}^{T} \mathbf{E}\left[f_t(x_t) - \hat{f}_t(x_t)\right] \tag{10}$$

$$- \sum_{t=1}^{T} \mathbf{E}\left[f_t(w) - \hat{f}_t(w)\right] \tag{11}$$

$$+ \sum_{t=1}^{T} \mathbf{E}\left[\hat{f}_t(x_t) - \hat{f}_t(w)\right] \tag{12}$$

where expectation is taken with respect to the player's choices, and $\hat{f}_t$ is defined as

$$\hat{f}_t(x) = \mathbf{E}[f_t(x + B_t v)], \quad \forall x \in \mathcal{K}$$

here $v \sim \mathbb{B}^n$ and the smoothing matrix $B_t$ is defined in Algorithm 1.

The sampling scheme used by Algorithm 1 yields an unbiased gradient estimate $g_t$ of the smoothed version $\hat{f}_t$, which is then inserted to FTARL-$\sigma$ (Algorithm 2). We can therefore interpret Algorithm 1 as performing noisy first-order method (FTARL-$\sigma$) over the smoothed versions. The $x_t$'s in Algorithm 1 are the outputs of FTARL-$\sigma$, thus the term in Equation (12) is associated with "exploitation". The other terms in Equations (9)-(11) measure the cost of sampling away from $x_t$, and the distance between the smoothed version and the original function, hence these term are associated with "exploration". In what follows we analyze these terms separately and show that Algorithm 1 achieves $\tilde{O}(\sqrt{T})$ regret.

**The Exploration Terms:** The next hold by the remark that follows Lemma 7 and by the lemma itself:

$$\mathbf{E}[f_t(y_t) - f_t(x_t)] = \mathbf{E}\left[\mathbf{E}_u[f_t(x_t + B_t u)] - f_t(x_t)\big|x_t]\right] \leq 0.5\beta \mathbf{E}\left[||B_t^2||_2\right] \leq \beta/2\eta\sigma t \tag{13}$$

$$- \mathbf{E}[f_t(w) - \hat{f}_t(w)] = \mathbf{E}\left[\mathbf{E}[\hat{f}_t(w) - f_t(w)\big|x_t]\right] \leq 0.5\beta \mathbf{E}\left[||B_t^2||_2\right] \leq \beta/2\eta\sigma t \tag{14}$$

$$\mathbf{E}[f_t(x_t) - \hat{f}_t(x_t)] = \mathbf{E}\left[\mathbf{E}[f_t(x_t) - \hat{f}_t(x_t)\big|x_t]\right] \leq 0 \tag{15}$$

where $||B_t^2||_2 \leq 1/\eta\sigma t$ follows by the definition of $B_t$ and by the fact that $\nabla^2 \mathcal{R}(x_t)$ is positive definite.

**The Exploitation Term:** The next Lemma bounds the regret of FTARL-$\sigma$ in the full-information setting:

**Lemma 11.** *Let $\mathcal{R}$ be a self-concordant barrier over a convex set $\mathcal{K}$, and $\eta > 0$. Consider an online player receiving $\sigma$-strongly-convex loss functions $h_1, \ldots, h_T$ and choosing points according to FTARL-$\sigma$ (Algorithm 2), and $\eta\|\nabla h_t(x_t)\|_t^* \leq 1/2, \ \forall t \in [T]$. Then the player's regret is upper bounded as follows:*

$$\sum_{t=1}^{T} h_t(x_t) - \sum_{t=1}^{T} h_t(w) \leq 2\eta \sum_{t=1}^{T} (\|\nabla h_t(x_t)\|_t^*)^2 + \eta^{-1}\mathcal{R}(w), \qquad \forall z \in \mathcal{K}$$

*here $(\|a\|_t^*)^2 = a^T(\nabla^2\mathcal{R}(x_t) + \eta\sigma t\mathcal{I})^{-1} a$*

Note that Algorithm 1 uses the estimates $g_t$ as inputs into FTARL-$\sigma$. Using Corollary 6 we can show that the $g_t$'s are unbiased estimates for the gradients of the smoothed versions $\hat{f}_t$'s. Using the regret bound of the above lemma, and the unbiasedness of the $g_t$'s, Lemma 9 ensures us:

$$\sum_{t=1}^{T} \mathbf{E}\left[\hat{f}_t(x_t) - \hat{f}_t(w)\right] \leq 2\eta \sum_{t=1}^{T} \mathbf{E}[(\|g_t\|_t^*)^2] + \eta^{-1}\mathcal{R}(w) \tag{16}$$

By the definitions of $g_t$ and $B_t$, and recalling $|f_t(x)| \leq L, \ \forall x \in \mathcal{K}$, we can bound:

$$\mathbf{E}[(\|g_t\|_t^*)^2|x_t] = \mathbf{E}\left[n^2\left(f_t(x_t + B_t u)\right)^2 u^\top B_t^{-1}\left(\nabla^2\mathcal{R}(x_t) + \eta\sigma t\mathcal{I}\right)^{-1} B_t^{-1} u \big| x_t\right] \leq (nL)^2$$

**Concluding:** Plugging the latter into Equation (16) and combining Equations (9)-(16) we get:

$$\mathbf{E}[\text{Regret}_T(w)] \leq 2\eta(nL)^2 T + \eta^{-1}\left(\mathcal{R}(w) + 2\beta\sigma^{-1}\log T\right) \tag{17}$$

Recall that $x_1 = \arg\min_{x\in\mathcal{K}} \mathcal{R}(x)$ and assume w.l.o.g. that $\mathcal{R}(x_1) = 0$ (we can always add $\mathcal{R}$ a constant). Thus, for a point $w \in \mathcal{K}$ such that $\pi_{x_1}(w) \leq 1 - T^{-1}$ Lemma 4 ensures us that $\mathcal{R}(w) \leq \nu \log T$. Combining the latter with Equation (17) and the choice of $\eta$ in Theorem 10 assures an expected regret bounded by $4nL\sqrt{(\nu + 2\beta\sigma^{-1})\,T\log T}$. For $w \in \mathcal{K}$ such that $\pi_{x_1}(w) > 1 - T^{-1}$ we can always find $w' \in \mathcal{K}$ such that $\|w - w'\| \leq \mathcal{O}(T^{-1})$ and $\pi_{x_1}(w') \leq 1 - T^{-1}$, using the Lipschitzness of the $f_t$'s, Theorem 10 holds.

**Correctness:** Note that Algorithm 1 chooses points from the set $\{x_t + \left(\nabla^2\mathcal{R}(x_t) + \eta\sigma t\mathcal{I}\right)^{-1/2} u, \ u \in \mathbb{S}^n\}$ which is inside the Dikin ellipsoid and therefore belongs to $\mathcal{K}$ (the Dikin Eliipsoid is always in $\mathcal{K}$). □

## 5 Summary and open questions

We have presented an efficient algorithm that attains near optimal regret for the setting of BCO with strongly-convex and smooth losses, advancing our understanding of optimal regret rates for bandit learning.

Perhaps the most important question in bandit learning remains the resolution of the attainable regret bounds for smooth but non-strongly-convex, or vice versa, and generally convex cost functions (see Table 1). Ideally, this should be accompanied by an efficient algorithm, although understanding the optimal rates up to polylogarithmic factors would be a significant advancement by itself.

**Acknowledgements**

The research leading to these results has received funding from the European Union's Seventh Framework Programme (FP7/2007-2013) under grant agreement n° 336078 – ERC-SUBLRN.

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
