[Reviews · NeurIPS 2014]

Submitted by Assigned_Reviewer_24

This paper details an approach to online bandit optimization for smooth strongly convex losses. This setting is in contrast to the online "full information" setting, where the gradient is made available to learner, instead feedback is received only through point-wise evaluations of the function. The authors describe a meta-algorithm that, through careful sampling strategies, constructs and unbiased approximation of the gradient and subsequently supplies it an optimization routine for the full information setting. The authors provide theoretical analysis for this approach and contrast the resulting bound O(T^{1/2}) with existing approaches.

The authors consider an important problem in online optimization, the paper is well written and each point is both clear and precise, and the final result is a clear improvement over existing methods. The proofs are straightforward and easy to follow and do not contain any obvious errors.

The paper hints at an interesting potential for future work, in the form of either an improved O(T^{1/2}) upper bound for smooth, or strictly convex losses or, alternatively, an \Omega(T^{2/3}) lower bound.

Summary: Authors present algorithm for online bandit optimization for smooth strictly convex losses with improved regret bound, paper is clear and concise, strong accept.

Submitted by Assigned_Reviewer_33

The authors consider the online convex optimization with bandit feedback. Until now we know rates of convergence in T^{2/3} when the loss functions are convex and either smooth or strongly convex . The authors make both assumption and prove convergence in T^{1/2}.

The paper is incremental. The algorithm is a mere adaptation of existing ones (Abernethy, Hazan & Rakhlin or Saha & Tewari) and the techniques of proof are almost exactly the same (basically, one inequality is improved using the strong convexity assumption).

The true interesting question would be whether we could remove any assumption (apart from Lipschitz) and still get T^{1/2}
Summary: This is another paper on online convex optimization with several assumptions on the loss functions. I find it is rather incremental.

Submitted by Assigned_Reviewer_40

In this paper, the authors have proposed a new algorithm for Bandit Convex Optimization (BCO) with strongly-convex and smooth loss functions, which uses an exploration scheme that shrinks with time. The authors have also proved that the proposed algorithm achieves an \tilde{O}(\sqrt{T}) regret (see Theorem 10), which matches the existing best lower bound if we ignore the logarithm factors. The technical analysis of this paper looks correct.

The result of this paper is significant in the sense that the authors have not only provided a tighter regret bound for BCO with strongly-convex and smooth loss functions (from \tilde{O}(T^{2/3}) to \tilde{O}(T^{1/2})), but more importantly, this improved regret bound matches the lower bound in (Shamir 2013). Thus, in general, I vote for acceptance of this paper. However, I vote for marginal acceptance instead of strong acceptance since I think the paper is not well-written, specifically,

1) There are many lemmas in the paper, but when reading the paper (without looking at the appendix), it is not clear which ones are existing results and which ones are proved by the authors. This makes it a little bit hard to evaluate the technical significance of the paper.

2) The authors spend too much space to review the existing results, and the main result of this paper starts at the end of page 6.

3) Some symbols in the paper are not well defined, for example, \delta in Eqn(4) is not defined. Some results are not very clear, for instance, the result of Lemma 9 should hold for any \omega, but the authors do not explicitly state that.

4) The authors should rewrite Algorithm 2: there should be no loop in Algorithm 2, and \Nabla h_t (x_t)'s should be inputs to the algorithm.

Summary: The result of the paper is significant, but the paper is not well-written and requires rewriting.
Author Feedback
Author rebuttal: We thank the reviewers for their time and effort, and appreciate their comments.

Reviewer 1:
===========

Many thanks for your positive review.

Reviewer 2:
===========

* novelty with respect previous work:
—> Our work proves a new bound on a fundamental class of functions, which was widely considered before, and gives the first tight regret bound for it, improving the exponent.
Technically, we have several new contributions over previous work, which include:
- We bound the distance between the smoothed function and the true function by the spectral norm of the smoothing matrix, and not only by a multiple of the diameter of the  convex body, as previously in Saha & Tewari.
-The ability to use an unbiased estimate of the gradient inside a first order method was proved by AHR only for the case of linear losses (this was also used by Saha & Tewari). In our case of nonlinear losses the analysis of AHR is not appropriate, and we had to extend a lemma from FKM.
- In our work the smoothed functions depend on the algorithm. We rigorously address this delicate feature.
-Previous works used sampling schemes which were time invariant, this technique was not sufficient to achieve our O(\sqrt{T}) bound, and we had to come up with a sampling scheme that shrinks with time

* “The true interesting question would be whether we could remove any assumption (apart from Lipschitz) and still get T^{1/2}”
—> This is absolutely correct. Getting O(\sqrt{T}) regret is perhaps the main open question in bandit learning today. We feel that our progress is a step in the right direction.

* "This is another paper on online convex optimization with several assumptions on the loss functions"
—> Strongly-convex and smooth functions, and specifically the quadratic loss, arise in numerous problems of both practical and theoretical importance. Moreover, these functions are extensively investigated in the literature of convex optimization. We believe that the a tight bound of O(\sqrt{T}) for the BCO setting with these functions will be of much interest for the online-learning community.

Reviewer 3:
===========
Thanks for your thorough review.

*"There are many lemmas in the paper, but when reading the paper (without looking at the appendix), it is not clear which ones are existing results and which ones are proved by the authors. This makes it a little bit hard to evaluate the technical significance of the paper ":
—> We made sure to reference all the lemmas in the paper. Specifically,
-Lemma 5: due to Flaxman Kalai & McMahan- we give a citation in line 187
-Corollary 6: due to by Saha & Tewari- we give a citation in line 215
-Lemma 9: generalization of a lemma from Flaxman Kalai & McMahan- we give a citation in line 256
The rest of the lemmas/theorems are new, to the best of our knowledge, including -Lemma 7, Theorem 10 and Lemma 11
-We will make sure to further emphasize these references in the final version.

*"The authors spend too much space to review the existing results, and the main result of this paper starts at the end of page 6":
—> In light of your comment we will shorten the review of previous work in the final version.

*"Some symbols in the paper are not well defined, for example, \delta in Eqn(4) is not defined. Some results are not very clear, for instance, the result of Lemma 9 should hold for any \omega, but the authors do not explicitly state that"
—> In the submitted version we have mistakenly omitted the following relation to \delta:
We should add “Let \delta>0” at the beginning of line 184.
This will be added to the final version.
—>Indeed lemma 9 holds for any \omega. Lemma 9 states: “Let \omega be a fixed point in K… “. We will further emphasize this point in the final version.

*”The authors should rewrite Algorithm 2: there should be no loop in Algorithm 2, and \Nabla h_t (x_t)'s should be inputs to the algorithm":
—> You are correct. In light of your comment we will simplify the presentation accordingly in the final version.